# Optimized Two-Port Laparoscopic-Assisted Ovariohysterectomy for Hydrometra and Pyometra in Small-Sized Dogs

**DOI:** 10.3390/ani15020187

**Published:** 2025-01-12

**Authors:** Young-Tae Park, Tomomi Minamoto

**Affiliations:** 1Ve. C. Jiyugaoka Animal Medical Center, Meguroku 152-0023, Japan; 2Evergreen Vet Research & Publication, Ichinomiya 491-0914, Japan; vetrandp@gmail.com

**Keywords:** laparoscopic surgery, laparoscopic-assisted surgery, ovariohysterectomy, hydrometra, pyometra, dogs, small breeds, two port

## Abstract

This study retrospectively investigated laparoscopic-assisted ovariohysterectomy, specifically for small-sized dogs weighing < 6 kg with hydrometra or pyometra. The present study aimed to investigate if a two-port laparoscopic approach would be safe and effective for these small-sized dogs. The findings showed that the surgery was successfully performed on 77 dogs, with a short recovery time and minimal complications. Most dogs were discharged the same day after surgery, and all recovered well. This technique could offer significant advantages, such as smaller surgical wounds, less postoperative pain, and quicker recovery. The results indicate that laparoscopic-assisted ovariohysterectomy could be a valuable option for treating mild to moderate uterine conditions in small-sized dogs.

## 1. Introduction

In dogs, hydrometra, mucometra, and pyometra are common diseases affecting the female reproductive system. While hydrometra and mucometra generally do not require urgent intervention, surgical removal of the uterus is often advised, due to their potential to progress to pyometra [1].

Clinical signs associated with pyometra include polydipsia, polyuria, and decreased appetite. In cases of closed pyometra, rapid progression to bacteremia and sepsis can occur, posing a serious threat to the animal’s life [2]. Although recent studies indicate that medical treatment provides a good long-term prognosis in approximately 86% of dogs with pyometra [3,4], surgical treatment remains the preferred approach, with a relatively low postoperative mortality rate of 1% [5].

In veterinary practice, laparoscopic ovariohysterectomy has recently been reported to offer significant advantages over open surgery, such as smaller incisions, reduced pain, and a lower rate of postoperative infections [6,7,8]. Techniques for laparoscopic-assisted ovariohysterectomy include single-port, two-port, three-port, and transvaginal methods [9,10,11]. Matsunami et al. reported that a three-port ovariohysterectomy was feasible in healthy dogs weighing < 5 kg [12]. Although one study recommends the two-port method due to its shorter operative time [13], no consensus has been reached regarding the superior technique. Recently, reduced-port laparoscopic surgery has become increasingly popular in human medicine due to its lower levels of surgical invasiveness [14,15]. The effectiveness of two-port laparoscopic adrenalectomy was reported in dogs [16]. Previous studies on laparoscopic ovariohysterectomy for treating pyometra included only medium-sized dogs (median weight > 20 kg) [17,18,19,20]. In small-sized dogs, performing laparoscopic surgery on an enlarged uterus can be challenging due to poor visualization and difficulty in maneuvering within the small abdominal cavity. Whether a two-port laparoscopic-assisted ovariohysterectomy is feasible in small-sized dogs with enlarged uteri remains unknown.

This study retrospectively investigated the clinical outcomes of a two-port laparoscopic ovariohysterectomy, combined with the use of a wound retractor and ultrasound probe cover in small-sized dogs (<6 kg) diagnosed with mild to moderate hydrometra or pyometra. We hypothesized that the laparoscopic-assisted removal of an enlarged uterus could be safely performed in small-sized dogs weighing < 6 kg, with clinical outcomes comparable to those reported for medium and large dogs.

## 2. Materials and Methods

### 2.1. Ethical Statement

Ethical review and approval were waived for this study, due to its retrospective design, with all procedures performed as part of routine clinical care. The surgeon informed all owners about the anesthetic and surgical procedures, as well as the associated risks. The owners consented to the surgery and the academic use of their patients’ data.

### 2.2. Study Design

The medical records of small-sized dogs (<6 kg) with an enlarged uterus due to hydrometra or pyometra that underwent laparoscopic-assisted ovariohysterectomy between 2018 and 2024 were reviewed retrospectively. The retrieved data included signalment, clinical signs, preoperative uterine horn diameter measured by ultrasound, operative time, length of hospital stay, intraoperative findings, intra- and postoperative complications, histopathological findings, and follow-up observations. Dogs suffering from perforation of the uterine horn, peritonitis, or significant uterine distension were considered unsuitable for laparoscopic-assisted ovariohysterectomy.

### 2.3. Pre-Anesthetic Protocol

Preoperative assessments included blood tests (i.e., complete blood count, chemistry panel, CRP, and coagulation tests), abdominal ultrasound, and thoracic and abdominal radiography. In all cases, food was withheld for at least 8 h prior to surgery. An intravenous catheter was inserted into the cephalic vein, and lactated Ringer’s solution was administered intravenously throughout the perioperative period.

### 2.4. Anesthesia Procedure

Midazolam (0.2 mg/kg), butorphanol (0.2 mg/kg), and atropine (25 µg/kg) were administered intravenously as pre-anesthetic medications. Propofol (6 mg/kg) was administered to induce anesthesia, followed by intubation of the dogs. After intubation, mechanical ventilation was applied, and 2.0−3.0% sevoflurane inhalation was used to maintain a surgical plane of anesthesia. If hypotension occurred intraoperatively, dopamine (constant rate infusion, 5−10 µg/kg/min), medetomidine (1 µg/kg), or ephedrine (0.1 mg/kg) was administered intravenously as necessary to stabilize blood pressure.

### 2.5. Two-Port Laparoscopic-Assisted Ovariohysterectomy

The dogs were positioned in dorsal recumbency, and the entire abdominal region, from the xiphoid process to the pubic bone, was shaved. A urinary catheter was then inserted. A 3 or 5 mm trocar was placed at the umbilicus using the modified Hasson technique. Carbon dioxide was insufflated into the abdominal cavity at a rate of 1.0–1.2 L/min to achieve an intra-abdominal pressure of 8 mmHg, establishing a pneumoperitoneum. After observing the entire abdominal cavity using a 30° oblique scope (HOPKINS, KARL STORZ Co., Ltd., Tokyo, Japan), a skin incision was made slightly caudal to the midpoint, between the umbilicus and the pelvic brim. This incision was approximately 5–10 mm larger than the maximum uterine horn diameter, and a 5 mm trocar (TERNAMIAN EndoTIP Cannula, KARL STORZ Co., Ltd., Tokyo, Japan) was placed under laparoscopic guidance. In cases where the preoperative uterine horn diameter exceeded 10 mm, or pyometra was suspected, an XXS or XS wound retractor (Smart Retractor, Top Co., Ltd., Tokyo, Japan) was placed at the two-port incision site to prevent contamination of the abdominal wall. A probe cover (Probe Cover G, Fuji Latex Co., Ltd., Tokyo, Japan) was used to cover the wound retractor (Figure 1).

The patient’s position was then adjusted from dorsal to right lateral recumbency. The surgeon stood on the ventral side of the animal, and the surgical table was tilted approximately 10° lower toward the surgeon’s side. An ultrasonic coagulation and cutting device (SONICBEAT, Olympus Co., Ltd., Tokyo, Japan) was inserted through the caudal port. The proper ligament of the left uterine horn was grasped using the coagulation and cutting device and retracted toward the abdominal wall to visualize the ovarian vessels and suspensory ligaments. A 3-0 or 4-0 polydioxanone suture with a round needle (1/2 circle, 17–22 mm) was inserted extracorporeally into the abdominal cavity using a needle holder, and the suture was placed around the proper ligament of the ovary as a stay suture to hold the uterine horn. Subsequently, the needle tip was placed within the abdominal wall (Figure 2).

The broad ligament of the uterus was coagulated and incised from the caudal to the cranial direction, using the ultrasonic coagulation and cutting device to separate the ovarian vessels and suspensory ligaments (Appendix A). After confirming the absence of bleeding or leakage of intrauterine fluid, the patient’s position was changed to left lateral recumbency, and the same procedure was performed on the right uterine horn. Using a trocar, the dissected right suspensory ligament was grasped and retracted externally from the caudal port. If a wound retractor and a probe cover had been used to create the second port, the probe cover was cut using scissors to remove the uterus from the body. When the second port was 5 mm in size, the abdominal wall was extended with scissors or a scalpel to safely exteriorize the uterus without rupturing the uterine horns. The cervical region was ligated using 3–0 or 2–0 polydioxanone sutures and transected using an ultrasonic surgical device (Appendix A). The uterine stump was sutured and returned to the abdominal cavity, followed by suturing the abdominal wall and subcutaneous tissue with a continuous pattern using polydioxanone sutures, and the skin incision was closed with a simple interrupted pattern using nylon sutures (Figure 3). After closing the caudal port, the pneumoperitoneum was re-established to check for bleeding or leakage of intrauterine fluid into the abdominal cavity. The umbilical camera port was closed in the same manner as the caudal port.

If the surgeon determined that the ovarian vessels and suspensory ligament could not be safely dissected due to inadequate visualization, the intra-abdominal pressure was increased to 10–12 mmHg for better visualization. If the procedure was still difficult, an additional 3–5 mm port was inserted at the midline between the xiphoid process and the umbilicus to manipulate the uterine tissues using grasping forceps. The procedure was converted to open surgery if adequate visualization could still not be achieved.

### 2.6. Postoperative Management

After the procedure was completed, bupivacaine (0.5%, 1 mg/kg) was administered at the incision site as an infiltration anesthetic, and meloxicam (0.2 mg/kg) was injected subcutaneously. Following recovery from anesthesia, the patient was monitored for approximately 5 h. For suspected cases of pyometra, oral amoxicillin (25 mg/kg BID) was prescribed for 1–2 weeks. All excised ovarian and uterine tissues were subjected to histopathological examination. Follow-up checkups for physical examination and observation of the surgical wound were performed in all dogs two weeks after surgery. Blood tests (e.g., white blood cell counts and CRP) were repeated for those with abnormal test results prior to surgery.

### 2.7. Statistical Analysis

The normality of the data distribution was assessed using the Shapiro–Wilk test. A parametric test (Student’s *t*-test) and a non-parametric test (Mann−Whitney *U* test) were applied as appropriate. A Chi-square test was used to test the homogeneity of breeds. Fisher’s exact test was used to analyze categorical data. Bonferroni correction was applied, with the significance level set at *p* < 0.007, to reduce the risk of a type I error.

## 3. Results

### 3.1. Retrieved Data

Seventy-seven dogs weighing ≤ 6 kg with hydrometra or pyometra were identified, as confirmed by abdominal ultrasonography. The median age was 8.8 years (range: 10 months–16.1 years) and the median weight was 3 (range: 1.26–6.0) kg. Fifty-one dogs with hydrometra were identified, comprising the following breeds: Toy poodles (n = 25), Chihuahuas (n = 10), mixed breeds (n = 7), Miniature dachshunds (n = 3), Pomeranians (n = 2), Miniature schnauzer (n = 1), French bulldog (n = 1), Shiba Inu (n = 1), and Maltese (n = 1). Seventy-seven dogs with pyometra were identified, comprising the following breeds: Chihuahuas (n = 7), Toy poodles (n = 6), Miniature Schnauzers (n = 3), Pomeranians (n = 3), mixed breeds (n = 3), Miniature dachshund (n = 1), Shih Tzu (n = 1), Cavalier King Charles spaniel (n = 1), and Yorkshire terrier (n = 1).

The clinical signs included lethargy, anorexia, polydipsia, and polyuria. Fifty-one (66%) dogs were asymptomatic, and an enlarged uterus was an incidental finding during routine health checkups. Histopathological examination of the uterus and ovaries revealed that 51 of the 77 dogs were diagnosed with hydrometra, while 26 were diagnosed with pyometra. The median maximum uterine horn diameter measured using ultrasonography was 10 (range: 4–30) mm. The median operative time, defined from the first skin incision to the wound closure, was 32 (range: 15–83) minutes, and the median length of hospital stay until discharge was 0 (range, 0–3) days. An expanded port with a wound retractor was used in all cases of pyometra and 12 cases of hydrometra. In four dogs, inadequate visualization during the two-port laparoscopic-assisted procedure necessitated additional measures; two cases required an additional port cranially on the midline, converting to a three-port laparoscopic-assisted procedure, while the remaining two cases were converted to open surgery.

No intraoperative complications were observed. Postoperative complications included decreased appetite within 3 days after surgery in 18 dogs and redness and swelling at the incision site indicative of infection in 3 dogs. The sutures were removed 7–10 days postoperatively in all dogs. At two-week follow-up checkups, all clinical signs had resolved after surgery, and all dogs were clinically well. The repeated blood tests showed that any abnormal blood test results were normalized.

### 3.2. Statistical Analysis Results

Table 1 summarizes signalments and other variables. Age and body weight did not differ between dogs with hydrometra and those with pyometra. The uterine horn sizes in dogs with pyometra (median [min–max] 15 [8–30] mm) were significantly larger than those in dogs with hydrometra (8.5 [4–25] mm, *p* < 0.007). There was no significant difference in operative time between the two groups. The length of hospital stay was significantly longer in dogs with pyometra (0 [0–3] days) compared to those with hydrometra (0 [0, 1] days, *p* < 0.007). The occurrence of surgical site infection did not differ between the two groups. However, the occurrence of postoperative anorexia was significantly higher in dogs with pyometra compared to those with hydrometra (*p* < 0.007).

## 4. Discussion

Laparoscopic surgery is increasingly being used in veterinary medicine due to its benefits, such as reduced surgical trauma, lower rates of surgical site infections, decreased inflammatory responses, and postoperative adhesions [6,8,21,22,23,24,25]. A study demonstrated that laparoscopic ovariohysterectomy was safe and feasible in clinically healthy dogs weighing < 5 kg [12]. In contrast, studies in humans indicate that smaller body sizes, such as those of newborns and infants, pose greater difficulty and limitations in the manipulation of laparoscopic instruments [26,27]. During laparoscopy, the surgical field is maintained by increasing the intra-abdominal pressure through CO_2_ insufflation. However, pressures exceeding 15 mmHg in dogs may lead to kidney damage [28], and CO_2_ insufflation during laparoscopic surgery can result in acute kidney injury in dogs with pre-existing chronic kidney disease [29]. Increased abdominal pressure increases the risk of complications due to changes in circulatory and respiratory system dynamics in dogs with pre-existing cardiovascular or respiratory disease [30]. Thus, thorough preoperative evaluations should be performed to exclude severe cardiovascular or respiratory cases, and abdominal pressure should be maintained as low as possible during the procedure. In the present study, laparoscopic procedures were performed on small-sized dogs weighing < 6 kg with an intra-abdominal pressure of 8 mmHg. Good visualization was maintained, allowing for the effective completion of procedures on most of the ovarian vessels and suspensory ligaments, despite the limited intra-abdominal space.

Devitt et al. described a two-port laparoscopic ovariectomy technique in which a suture needle was inserted externally, passed through the uterine horn, and then brought back outside the body to facilitate tissue traction [8]. However, another study reported the risk of needle breakage associated with this technique [31]. Based on these findings, we kept the needle tip within the abdominal wall after suturing the proper ovarian ligament, minimizing the chance of complications. This approach was easily achieved because the tip of the needle was easily visualized due to the thin abdominal wall of smaller dogs.

In the present study, the ovarian vessels and suspensory ligaments were dissected successfully without intraoperative hemorrhage in all cases. Ultrasonic coagulation and cutting devices provide a rapid and safe method for dissecting vessels of up to 3–5 mm in diameter [21,32]. Clipping or suture ligation has traditionally been used for vessel and ligament dissection, but in small-sized dogs, ultrasonic coagulation or vessel sealing devices offer an advantage because of their smaller vessel sizes, contributing to reduced surgery time [33,34,35]. Additionally, the use of ultrasonic coagulation and cutting devices for ovarian vessel dissection results in less postoperative pain compared to suture ligation [21,36]. Although the vessels in dogs with pyometra may be dilated due to inflammation, ultrasonic coagulation and cutting devices effectively managed vessel dissection in the present study. Previous studies reported median operative times of 85 min, 57 min, and 107 min for single-port, two-port, and three-port methods for pyometra, respectively [18,19,20], while the present study achieved a median operative time of 32 min. Although the intra-abdominal space was limited, the reduced tissue size in small-sized dogs likely contributed to the shorter operative time.

The mortality rate after conventional open surgery for the treatment of pyometra in dogs is reported to be 3–20%, with a complication rate of 20%, of which sepsis accounted for 12% [4]. Conversion to laparotomy was carried out due to uterine rupture in 8.3% of cases during three-port laparoscopic ovariohysterectomy [20] and 14% during single-port surgery [19]. A study on two-port surgery using a surgical glove port reported no instances of uterine rupture or conversion to laparotomy [18]. The complication rate in the present study was better than in the previously reported studies. Conversion to laparotomy was carried out due to poor visibility of the surgical field in 2.5% of cases, and an additional port was necessary in 2.5% of cases. No leakage of uterine contents into the abdominal cavity was observed in the present study. Meticulous manipulation of the proper ovarian ligament minimized the risk of uterine lumen perforation. In four cases, the two-port surgery was not completed because the uterine horn near the ovary was distended, impairing the visibility of the ovarian vessels. Tilting the surgical table toward the surgeon during laparoscopic-assisted ovariohysterectomy can improve the visibility of the surgical field [37]; however, due to gravity, the uterine horn may tilt inward, potentially limiting the visibility of the ovarian vessels. Adding an operational port can improve the grasping of the uterine horn, enhancing visibility. However, if visibility remains compromised, conversion to laparotomy is necessary. Generally, surgically treated pyometra cases can be discharged within 1−2 days if no complications occur [4]; however, in the present study, most cases were discharged on the same day.

Even though the uterine horn in pyometra cases was significantly distended compared to in dogs with hydrometra, surgery was completed in a timely manner for both conditions. The occurrence of surgical site infections did not differ between pyometra and hydrometra cases. However, the occurrence of postoperative anorexia and duration of hospitalization were higher in pyometra cases compared to hydrometra cases, indicating that postoperative medical management is necessary even with minimally invasive surgical intervention. On the other hand, most hydrometra cases were discharged on the same day as the surgery, suggesting that a two-port laparoscopic-assisted ovariohysterectomy should be indicated for hydrometra cases.

A wound retractor is recommended for wounds > 10 mm to prevent infection or tumor dissemination in the abdominal wall [38]. The probe cover port method, which involves covering a wound retractor with a surgical glove, offers a cost-effective alternative to traditional single-port devices [39]. This method has been shown to reduce operative time for canine ovariohysterectomy [40] and is effective in treating pyometra. In the present study, because all patients required a wound incision of least 10 mm to exteriorize the uterus, wound retractors were used proactively to protect the abdominal wall. XXS- or XS-sized wound retractors were used in all cases. The commonly available wrist diameter of Size 5 surgical gloves is approximately 67 mm, which does not completely seal an XXS-sized wound retractor outer ring (65 mm), potentially causing pneumoperitoneum leakage. Therefore, a 35 mm-diameter ultrasound probe cover made of sterilized rubber was placed over the outer rings of the XXS and XS wound retractors, and pneumoperitoneum leakage did not occur. Although an ultrasound probe cover can hold only a single trocar, as opposed to five in a surgical glove, its semi-transparent nature aids forceps manipulation at a reduced cost. This method may provide a new option for placing trocars in smaller wound retractors.

In cases of hydrometra, the uterine contents are not bacteria-contaminated fluid [2]; therefore, using a wound retractor is not always necessary. Instead, the caudal port can be extended to allow for the exteriorization of the uterus. Ideally, the use of a wound retractor should be determined preoperatively based on the size of the fluid-filled uterine horn, as measured by ultrasonography, and the presence of pyometra.

The limitations of this study include its retrospective nature. Only dogs with mild to moderate pyometra were considered for this surgery. Those with perforations, peritonitis, or significant uterine distension were excluded. In such cases, open ovariohysterectomy should be indicated. Finally, only one surgeon performed all the surgeries. In addition, the ultrasound probe cover for the wound retractor may not be readily available in some clinical settings. Bonferroni correction was used to prevent type I errors of multiple comparisons. However, it can potentially increase the risk of type II errors and decrease statistical power. The retrospective nature of the study and its limited sample size necessitate further investigation. Future studies with larger case numbers are warranted to investigate the clinical outcomes of two-port laparoscopic-assisted ovariohysterectomy for mild to moderate hydrometra and pyometra.

## 5. Conclusions

Two-port laparoscopic-assisted ovariohysterectomy can be performed safely and effectively in dogs weighing < 6 kg with mild to moderate hydrometra or pyometra.

## Figures and Tables

**Figure 1 animals-15-00187-f001:**
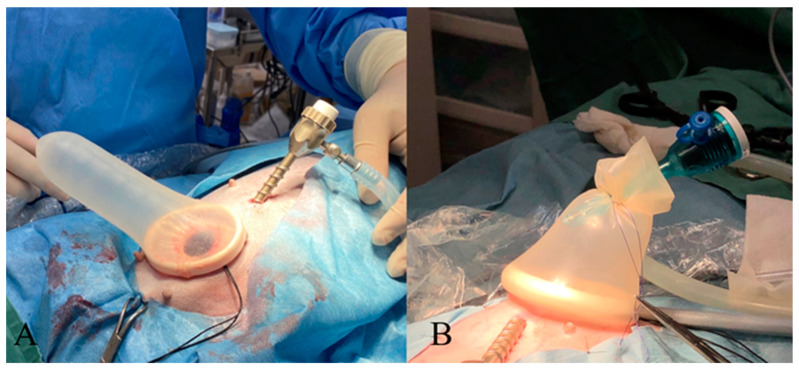
(**A**) A probe cover is placed over a wound retractor. (**B**) A cannula is placed inside the probe cover, serving as a probe cover port. The cannula is secured using a 3-0 nylon suture.

**Figure 2 animals-15-00187-f002:**
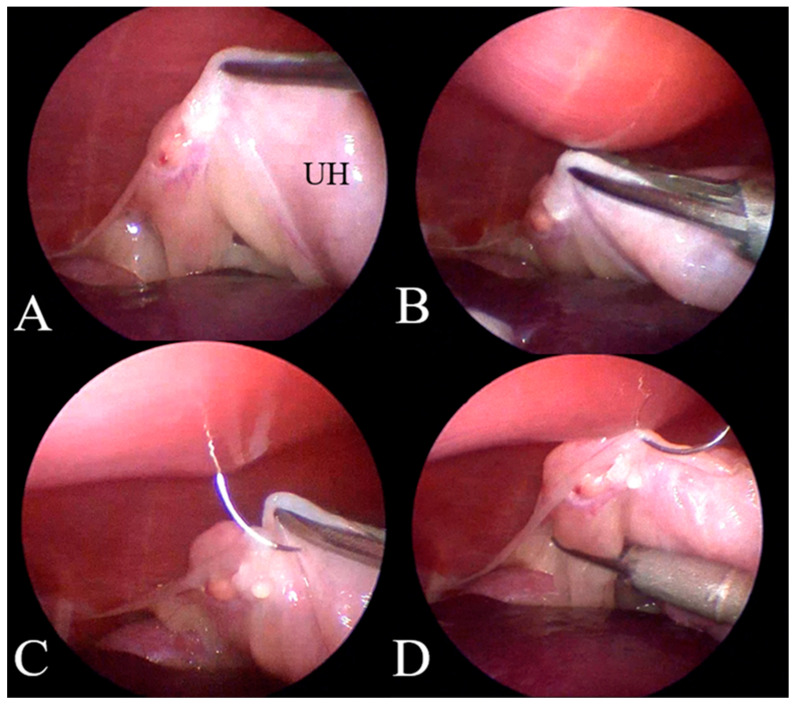
(**A**) The distended uterine horn is elevated by holding the suspensory ligament. UH: Uterus Horn. (**B**) The proper suturing site on the abdominal wall is determined by externally pushing the abdominal wall with a finger. (**C**) A round needle suture is inserted extracorporeally into the abdominal cavity using a needle holder, with the suture placed around the proper ligament of the ovary as a stay suture to hold the uterine horn. (**D**) While holding the stay suture, the ovarian artery, vein, and ovarian ligament are dissected using an ultrasonic coagulation device. The needle tip remains within the abdominal wall.

**Figure 3 animals-15-00187-f003:**
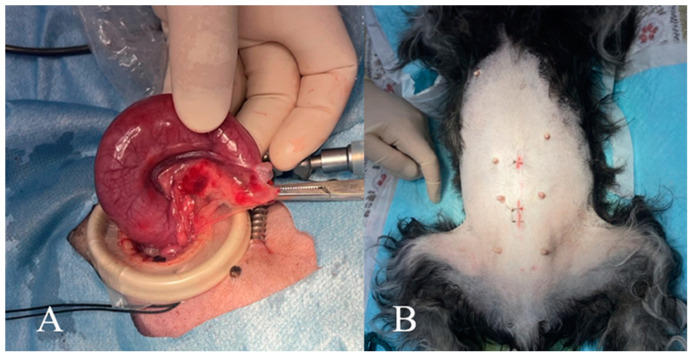
(**A**) The enlarged uterus is removed. (**B**) Surgical wounds are shown.

**Table 1 animals-15-00187-t001:** Summary of variables in dogs with hydrometra and pyometra.

Variables	Hydrometra (n = 51)	Pyometra (n = 26)	*p*-Value
Age (months)	105 (10–194)	122 (39–192)	*p* = 0.024
Body weight (kg)	3 (1.26–6.0)	3 (1.6–6.0)	*p* = 0.539
Uterine horn size (mm)	8.5 (4–25)	15 (8–30)	*p* < 0.007
Surgery time (min)	31 (15–83)	33.5 (15–64)	*p* = 0.34
Postoperative anorexia	3	15	*p* < 0.007
Surgical site infection	3	0	*p* = 0.547
Hospital stay (days)	0 (0–1)	0 (0–3)	*p* < 0.007

The data are presented as median (minimum−maximum) values. A *p*-value < 0.007 was considered statistically significant.

## Data Availability

The dataset is available upon request from the authors.

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
