# Peer review of "Optimized Two-Port Laparoscopic-Assisted Ovariohysterectomy for Hydrometra and Pyometra in Small-Sized Dogs"

_animals, 2025, doi:10.3390/ani15020187_

Round 1
Reviewer 1 Report
Comments and Suggestions for Authors
Dear Authors,
Thank you for your submission in our journal.
I would like to express my gratitude for your dedication in preparing and submitting this manuscript regarding laparoscopic-assisted ovariohysterectomy in small dogs. This study can addresses a relevant and timely topic in veterinary surgery, highlighting a minimally invasive approach that could provide important information for both practitioners and researchers.
Nonetheless, I have several concerns regarding your techniques and evaluation, which influence the paper's scientific soundness and clarity. Endorsing the paper in its current state remains difficult. To help you in enhancing your submission, I've provided a detailed critique below, complete with suggestions that could improve the paper substantially for resubmission.
Best Regards
Review of Manuscript
The study presents an interesting investigation into laparoscopic-assisted ovariohysterectomy in small dogs, highlighting the benefits of minimally invasive techniques. However, several methodological and analytical shortcomings must be addressed to strengthen its scientific validity and provide a solid basis for resubmission.
Materials and Method
Lack of Statistical Information
The absence of detailed statistical methods undermines the rigor of the study. Key aspects that are missing include defined inclusion and exclusion criteria, handling of clinical and demographic variables, statistical methods for analysis such as comparative tests or p-values, and justification for sample size.
Recommendation: Add a dedicated subsection addressing data management and statistical analysis. Comparing outcomes between hydrometra and pyometra cases or surgical complications could provide valuable insights.
Recommendation: Reorganize into distinct subsections: Study Population specifying criteria and timeframe, Surgical and Anesthetic Protocols providing a concise overview, and Data Collection and Outcomes emphasizing complications, operative times, and follow-up.
Limited Focus on Clinical Outcomes
The lack of clear primary and secondary outcomes (e.g., success rates, complications, recovery times) diminishes the study's relevance.
Recommendation: Clearly define these outcomes and describe methods for their assessment.
Retrospective Nature and Bias
The study does not adequately address potential biases inherent in retrospective designs, such as operator variability or case selection.
Recommendation: Discuss these limitations and describe measures to mitigate bias where applicable.
Ethical and Authorization Aspects
No mention is made of ethical approval or informed consent, which is critical even in retrospective studies.
Recommendation: Include a statement confirming ethical approval and data management protocols.
Results
Lack of Statistical Data
The results section lacks statistical comparisons and measures of significance, limiting the ability to draw robust conclusions.
Recommendation: Include statistical analyses (e.g., comparisons by condition or breed) and report p-values to enhance interpretability.
Data Presentation
While descriptive statistics and figures are present, their value is diminished by the lack of statistical context.
Recommendation: Supplement with inferential statistics to provide meaningful insights.
Content Misplacement
Descriptions of surgical techniques belong in the Materials and Methods section, not in the results.
Recommendation: Relocate technical details to the appropriate section.
Postoperative Complications
Complications are mentioned but not analyzed in relation to clinical variables or preoperative data.
Recommendation: Correlate complications with relevant variables to enhance the clinical impact of this section.
Discussion
The discussion highlights the benefits of laparoscopy but lacks depth in several areas.
Limited Consideration of Clinical Variables
The influence of factors such as age, health status, and comorbidities is not discussed.
Recommendation: Analyze the impact of these variables on outcomes and complications.
Insufflation Safety
The safety of COâ‚‚ insufflation is mentioned without sufficient context regarding pre-existing conditions or higher pressures.
Recommendation: Incorporate references or data from previous studies on insufflation safety in similar populations.
Operative Results and Recovery
While operative time is noted, no analysis of recovery times or postoperative quality of life is provided.
Recommendation: Compare these factors with alternative methods to contextualize the technique's effectiveness.
Postoperative Monitoring
There is no discussion of strategies for assessing complications postoperatively.
Recommendation: Outline a follow-up plan for monitoring long-term outcomes.
Sample Size and Generalizability
The limited sample size and exclusion of complex cases reduce generalizability.
Recommendation: Emphasize the need for larger, more diverse samples in future studies.
Comparison with Alternative Techniques
The manuscript does not adequately compare the proposed technique with alternatives. Discuss the advantages and disadvantages of other methods supported by relevant evidence (bibliography.
Ethical Considerations
Adherence to ethical guidelines is not mentioned.
Recommendation: Include a statement detailing ethical approvals and animal welfare protocols.
Critical Conclusion
While the study addresses an important topic in veterinary surgery, the methodological and analytical gaps substantially limit its scientific validity. A thorough revision addressing these issues is necessary for future consideration. The proposed revisions aim to guide the authors in enhancing the clarity, rigor, and clinical relevance of the manuscript.
Author Response
Reviewer 1
Thanks for reviewing our manuscript .
I have made corrections based on your suggestions.
Please kindly review the revised version.
- Materials and Method
Lack of Statistical Information
The absence of detailed statistical methods undermines the rigor of the study. Key aspects that are missing include defined inclusion and exclusion criteria, handling of clinical and demographic variables, statistical methods for analysis such as comparative tests or p-values, and justification for sample size.
Recommendation: Add a dedicated subsection addressing data management and statistical analysis. Comparing outcomes between hydrometra and pyometra cases or surgical complications could provide valuable insights.
Author’s response: Thank you for your suggestion. We performed statistical analyses to compare the signalment and variables between hydrometra and pyometra cases.
Line 163-168:
2.8. Statistical Analysis
The normality of the data distribution was assessed using the Shapiro-Wilk test. A parametric test (Student’s t-test) and a non-parametric test (Mann−Whitney U test) were applied as appropriate. A Chi-square test was used to test the homogeneity of breeds. Fisher’s exact test was used to analyze categorical data. Bonferroni correction was applied, with the significance level set at p < 0.007.
Line 223-232
3.2. Statistical Analysis Results
Table 1 summarizes signalments and other variables. Age and body weight did not differ between dogs with hydrometra and those with pyometra. The uterine horn sizes in dogs with pyometra (median [min−max] 15 [8–30] mm) were significantly larger than those in dogs with hydrometra (8.5 [4–25] mm, p < 0.007). There was no significant difference in operative time between the two groups. The length of hospital stay was significantly longer in dogs with pyometra (0 [0–3] days) compared to those with hydrometra (0 [0–1] days, p < 0.007). The occurrence of surgical site infection did not differ between the two groups. However, the occurrence of postoperative anorexia was significantly higher in dogs with pyometra compared to those with hydrometra (p < 0.007).
- Recommendation: Reorganize into distinct subsections: Study Population specifying criteria and timeframe, Surgical and Anesthetic Protocols providing a concise overview, and Data Collection and Outcomes emphasizing complications, operative times, and follow-up.
Author’s response: We organized the Materials and Methods section and Results section using distinct subheadings.
- Materials and Methods subheadings: ethical statement, study design, preanesthetic protocol, anesthesia procedure, two-port laparoscopic-assisted ovariohysterectomy, postoperative management, statistical analysis, figures
- Results subheadings: retrieved data, statistical results, tables
- Limited Focus on Clinical Outcomes
The lack of clear primary and secondary outcomes (e.g., success rates, complications, recovery times) diminishes the study's relevance.
Recommendation: Clearly define these outcomes and describe methods for their assessment.
Author’s response: We compared signalments, uterine horn size, surgery time, postoperative anorexia, surgical infection, and hospital stays between hydrometra and pyometra cases. These results are summarized in Table 1.
Table 1. Summary of variables in dogs with hydrometra and pyometra
Variables |
Hydrometra (n = 51) |
Pyometra (n = 26) |
p-value |
Age (months) |
105 (10–194) |
122 (39–192) |
p = 0.024 |
Body weight (kg) |
3 (1.26–6.0) |
3 (1.6–6.0) |
p = 0.539 |
Uterine horn size (mm) |
8.5 (4–25) |
15 (8–30) |
p < 0.007 |
Surgery time (min) |
31 (15–83) |
33.5 (15–64) |
p = 0.34 |
Postoperative anorexia |
3 |
15 |
p < 0.007 |
Surgical site infection |
3 |
0 |
p = 0.547 |
Hospital stays (days) |
0 (0–1) |
0 (0–3) |
p < 0.007 |
The data are presented as median (minimum−maximum) values. A p-value < 0.007 was considered statistically significant.
Retrospective Nature and Bias
The study does not adequately address potential biases inherent in retrospective designs, such as operator variability or case selection.
- Recommendation: Discuss these limitations and describe measures to mitigate bias where applicable.
Author’s response: We included the sentences regarding the potential biases of a retrospective study as a limitation in the revised manuscript.
Line 320-324
The limitations of this study include its retrospective nature. Only dogs with mild pyometra were considered for this surgery. Those with perforations, peritonitis, or significant uterine distension were excluded. Finally, only one surgeon performed all the surgeries.
- Ethical and Authorization Aspects
No mention is made of ethical approval or informed consent, which is critical even in retrospective studies.
Recommendation: Include a statement confirming ethical approval and data management protocols.
Author’s response: We included a statement regarding the ethical statement and data management.
Line 69-73
2.1. Ethical Statement
Ethical review and approval were waived for this study due to its retrospective design, with all procedures performed as part of routine clinical care. The surgeon informed all owners about the anesthetic and surgical procedures, as well as the associated risks. The owners consented to the surgery and the academic use of their patients’ data.
- Results
Lack of Statistical Data
The results section lacks statistical comparisons and measures of significance, limiting the ability to draw robust conclusions.
Recommendation: Include statistical analyses (e.g., comparisons by condition or breed) and report p-values to enhance interpretability.
Author’s response: We performed statistical analyses to address this issue.
Please refer to our answers to your questions #1 and 3.
- Data Presentation
While descriptive statistics and figures are present, their value is diminished by the lack of statistical context.
Recommendation: Supplement with inferential statistics to provide meaningful insights.
Author’s response: We performed statistical analyses to address this issue.
- Content Misplacement
Descriptions of surgical techniques belong in the Materials and Methods section, not in the results.
Recommendation: Relocate technical details to the appropriate section.
Author’s response: We understood that this comment was about the figure images for the surgical procedure. We relocated them to the M&M section. If this does not address your concern, please let us know.
- Postoperative Complications
Complications are mentioned but not analyzed in relation to clinical variables or preoperative data.
Recommendation: Correlate complications with relevant variables to enhance the clinical impact of this section.
Author’s response: Thank you for your suggestion. Since the postoperative complications were minor (i.e., anorexia and surgical site infection) and resolved within a few days, and a follow-up period was short, we concluded that correlation analysis was not necessary for this study.
- Discussion
The discussion highlights the benefits of laparoscopy but lacks depth in several areas.
Limited Consideration of Clinical Variables
The influence of factors such as age, health status, and comorbidities is not discussed.
Recommendation: Analyze the impact of these variables on outcomes and complications.
Author’s response: Since the postoperative complications were minor (i.e., anorexia and surgical site infection) and resolved within a few days, and clinical outcomes were good, we concluded that analyzing the impact of the variables on outcomes and complications was not necessary for this study.
- Insufflation Safety
The safety of COâ‚‚ insufflation is mentioned without sufficient context regarding pre-existing conditions or higher pressures.
Recommendation: Incorporate references or data from previous studies on insufflation safety in similar populations.
Author’s response: We incorporated the references from previous studies on insufflation safety.
Line 245-252
However, pressures exceeding 15 mmHg in dogs may lead to kidney damage [28], and CO2 insufflation during laparoscopic surgery can result in acute kidney injury in dogs with pre-existing chronic kidney disease [29]. Increased abdominal pressure increases the risk of complications due to changes in circulatory and respiratory system dynamics in dogs with pre-existing cardiovascular or respiratory disease [30]. Thus, thorough preoperative evaluations should be performed to exclude severe cardiovascular or respiratory cases, and abdominal pressure should be maintained as low as possible during the procedure.
- Operative Results and Recovery
While operative time is noted, no analysis of recovery times or postoperative quality of life is provided.
Recommendation: Compare these factors with alternative methods to contextualize the technique's effectiveness.
Author’s response:
Line 274-288 Previous studies reported median operative times of 85 minutes, 57 minutes, and 107 minutes for single-port, two-port, and three-port methods for pyometra, respectively [18–20], while the present study achieved a median operative time of 32 minutes. Although the intra-abdominal space was limited, the reduced tissue size in small-sized dogs likely contributed to the shorter operative time. The mortality rate after conventional open surgery for the treatment of pyometra in dogs is reported to be 3–20%, with a complication rate of 20%, of which sepsis accounted for 12% [4]. Conversion to laparotomy was made due to uterine rupture in 8.3% of cases during three-port laparoscopic ovariohysterectomy [20] and 14% during single-port surgery [19]. A study on two-port surgery using a surgical glove port reported no instances of uterine rupture and conversion to laparotomy [18]. The complication rate in the present study was better than the previously reported studies. Conversion to laparotomy was made due to poor visibility of the surgical field in 2.5% of cases, and an additional port was necessary in 2.5% of cases. No leakage of uterine contents into the abdominal cavity was observed in the present study.
- Postoperative Monitoring
There is no discussion of strategies for assessing complications postoperatively.
Recommendation: Outline a follow-up plan for monitoring long-term outcomes.
Author’s response: We included a follow-up protocol in the M&M section. All cases recovered well 2 weeks after surgery, and abnormal blood test results improved. Thus, we did not monitor long-term outcomes.
- Sample Size and Generalizability
The limited sample size and exclusion of complex cases reduce generalizability.
Recommendation: Emphasize the need for larger, more diverse samples in future studies.
Author’s response: We mentioned this as a study limitation in the discussion.
Line 324-327 The retrospective nature of the study and its limited sample size necessitate further investigation. Future studies with larger case numbers are warranted to investigate the clinical outcomes of two-port laparoscopic-assisted ovariohysterectomy for hydrometra and pyometra.
- Comparison with Alternative Techniques
The manuscript does not adequately compare the proposed technique with alternatives. Discuss the advantages and disadvantages of other methods supported by relevant evidence (bibliography.
Author’s response: Thank you for your suggestion. We compared our technique with previously reported techniques in the revised manuscript.
Line: 274-288
Previous studies reported median operative times of 85 minutes, 57 minutes, and 107 minutes for single-port, two-port, and three-port methods for pyometra, respectively [18–20], while the present study achieved a median operative time of 32 minutes. Although the intra-abdominal space was limited, the reduced tissue size in small-sized dogs likely contributed to the shorter operative time. The mortality rate after conventional open surgery for the treatment of pyometra in dogs is reported to be 3–20%, with a complication rate of 20%, of which sepsis accounted for 12% [4]. Conversion to laparotomy was made due to uterine rupture in 8.3% of cases during three-port laparoscopic ovariohysterectomy [20] and 14% during single-port surgery [19]. A study on two-port surgery using a surgical glove port reported no instances of uterine rupture and conversion to laparotomy [18]. The complication rate in the present study was better than the previously reported studies. Conversion to laparotomy was made due to poor visibility of the surgical field in 2.5% of cases, and an additional port was necessary in 2.5% of cases. No leakage of uterine contents into the abdominal cavity was observed in the present study.
- Ethical Considerations
Adherence to ethical guidelines is not mentioned.
Recommendation: Include a statement detailing ethical approvals and animal welfare protocols.
Author’s response: We included a statement regarding the ethical statement.
Line 69-73
2.1. Ethical Statement
Ethical review and approval were waived for this study due to its retrospective design, with all procedures performed as part of routine clinical care. The surgeon informed all owners about the anesthetic and surgical procedures, as well as the associated risks. The owners consented to the surgery and the academic use of their patients’ data.
Reviewer 2 Report
Comments and Suggestions for Authors
Revised Study Highlights
The present study demonstrates the optimization of the Two-Port Laparoscopic-Assisted Ovariohysterectomy technique in two small dogs diagnosed with hydrometra or pyometra. This study highlights a minimally invasive surgical approach that can aid veterinarians in performing abdominal surgeries more effectively.
While the data presented is sufficient for potential publication, substantial revision is required to enhance its content and presentation.
Suggestions for Different Sections
Title Suggestion
The current title, "Two-Port Laparoscopic-Assisted Ovariohysterectomy in Small 2 Dogs with Hydrometra or Pyometra," lacks a catchy appeal. Consider revising it to something more engaging and concise, such as:
"Optimized Two-Port Laparoscopic-Assisted Ovariohysterectomy for Hydrometra and Pyometra in Small Dogs."
Introduction
Provide a broader context on the Two-Port Laparoscopic technique, including its application in other abdominal surgeries in dogs. Discuss its advantages over traditional methods, focusing on reduced invasiveness, shorter recovery times, and fewer complications.
Materials and Methods
Ethical Statement
Include a clear ethical statement ensuring that the study adheres to relevant ethical guidelines for animal research.
Structural Improvements
Organize the section with distinct subheadings to improve readability and flow:
Ethical Statement
Dog Breeds Included
Inclusion and Exclusion Criteria
Pre-operative Protocol
Anesthesia/Medication Details
Laparoscopic Technique Description
Post-operative Management
Results
Shift the dog breed information from this section to the Materials and Methods section.
Include a table summarizing key data (mean values and ranges) for the studied parameters:
Total number of cases
Cases with hydrometra and pyometra
Uterine horn size
Incision size
Surgery duration
Healing time
Percentage of cases with post-operative complications
Consider presenting the surgical technique in a video format to make the procedure easier to follow.
Discussion
The discussion effectively highlights the advantages of the technique and offers valuable tips for managing complications during the procedure. Maintain this section while ensuring logical flow and coherence with the study's findings.
Conclusion
The conclusion is satisfactory and requires no significant changes.
Author Response
Reviewer 2
Thanks for reviewing my manuscript.
I have made corrections based on your suggestions.
Please kindly review the revised version.
- Title Suggestion
The current title, "Two-Port Laparoscopic-Assisted Ovariohysterectomy in Small 2 Dogs with Hydrometra or Pyometra,"lacks a catchy appeal. Consider revising it to something more engaging and concise, such as:
"Optimized Two-Port Laparoscopic-Assisted Ovariohysterectomy for Hydrometra and Pyometra in Small Dogs."
Author’s response: Thank you for your suggestion. We changed our title accordingly. Line 2-3 “Optimized Two-Port Laparoscopic-Assisted Ovariohysterectomy for Hydrometra and Pyometra in Small-Sized Dogs”
- Introduction
Provide a broader context on the Two-Port Laparoscopic technique, including its application in other abdominal surgeries in dogs. Discuss its advantages over traditional methods, focusing on reduced invasiveness, shorter recovery times, and fewer complications.
Author’s response: We added a new reference and discussed the advantages of the two-port laparoscopic technique.
Line 52-56 Although one study recommends the two-port method due to its shorter operative time [13], no consensus has been reached regarding the superior technique. Recently, reduced-port laparoscopic surgery has become increasingly popular due to lesser surgical invasiveness in human medicine. The effectiveness of two-port laparoscopic adrenalectomy was reported in dogs.
Materials and Methods
- Ethical Statement
Include a clear ethical statement ensuring that the study adheres to relevant ethical guidelines for animal research.
Author’s response: We included the information regarding the relevant ethical guidelines, including owner consent procedure.
Line 69-73
2.1. Ethical Statement
Ethical review and approval were waived for this study due to its retrospective design, with all procedures performed as part of routine clinical care. The surgeon informed all owners about the anesthetic and surgical procedures, as well as the associated risks. The owners consented to the surgery and the academic use of their patients’ data.
- Structural Improvements
Organize the section with distinct subheadings to improve readability and flow:
Ethical Statement
Dog Breeds Included
Inclusion and Exclusion Criteria
Pre-operative Protocol
Anesthesia/Medication Details
Laparoscopic Technique Description
Post-operative Management
Author’s response: We organized the Materials & Methods section with the subheadings. Line 68 -168
- Results
Shift the dog breed information from this section to the Materials and Methods section.
Author’s response: Since this is a retrospective study, we think that this information should be included in the result section. However, please let me know if you still recommend this be in the Materials and Methods section.
- Include a table summarizing key data (mean values and ranges) for the studied parameters:
Total number of cases
Cases with hydrometra and pyometra
Uterine horn size
Incision size
Surgery duration
Healing time
Percentage of cases with post-operative complications
Author’s response: We summarized the data in a table (Table 1). We also performed statistical analyses to compare the parameters between dogs with hydrometra and dogs with pyometra and included the p-values in the table. Unfortunately, we do not have data regarding the incision size for all dogs.
Line 223-232 3.2. Statistical Analysis Results
Table 1 summarizes signalments and other variables. Age and body weight did not differ between dogs with hydrometra and those with pyometra. The uterine horn sizes in dogs with pyometra (median [min−max] 15 [8–30] mm) were significantly larger than those in dogs with hydrometra (8.5 [4–25] mm, p < 0.007). There was no significant difference in operative time between the two groups. The length of hospital stay was significantly longer in dogs with pyometra (0 [0–3] days) compared to those with hydrometra (0 [0–1] days, p < 0.007). The occurrence of surgical site infection did not differ between the two groups. However, the occurrence of postoperative anorexia was significantly higher in dogs with pyometra compared to those with hydrometra (p < 0.007).
Table 1. Summary of variables in dogs with hydrometra and pyometra
Variables |
Hydrometra (n = 51) |
Pyometra (n = 26) |
p-value |
Age (months) |
105 (10–194) |
122 (39–192) |
p = 0.024 |
Body weight (kg) |
3 (1.26–6.0) |
3 (1.6–6.0) |
p = 0.539 |
Uterine horn size (mm) |
8.5 (4–25) |
15 (8–30) |
p < 0.007 |
Surgery time (min) |
31 (15–83) |
33.5 (15–64) |
p = 0.34 |
Postoperative anorexia |
3 |
15 |
p < 0.007 |
Surgical site infection |
3 |
0 |
p = 0.547 |
Hospital stays (days) |
0 (0–1) |
0 (0–3) |
p < 0.007 |
The data are presented as median (minimum−maximum) values. A p-value < 0.007 was considered statistically significant.
- Consider presenting the surgical technique in a video formatto make the procedure easier to follow.
Author’s response: Thank you for your suggestion. We submitted a video for the procedure as a supplementary file.
- Discussion
The discussion effectively highlights the advantages of the technique and offers valuable tips for managing complications during the procedure. Maintain this section while ensuring logical flow and coherence with the study's findings.
Author’s response: Thank you for your comment. We revised the discussion section for better clarity and readability while ensuring logical flow.
- Conclusion
The conclusion is satisfactory and requires no significant changes.
Author’s response: Thank you for your comment.
Round 2
Reviewer 1 Report
Comments and Suggestions for Authors
Dear Authors,
Thank you for submitting your manuscript to our journal. We appreciate the effort and dedication that went into this study, which addresses an innovative and clinically relevant topic—the application of two-port laparoscopic-assisted ovariohysterectomy in small-sized dogs. The detailed methodology and clear presentation of results are commendable, and the study has the potential to make a significant contribution to veterinary surgical practices.
After thoroughly reviewing the manuscript, we have identified several strengths, including the reproducibility of the described methods and the clinical relevance of the findings. However, some areas require further clarification and refinement to meet the journal's standards for publication.
In particular, we suggest that you provide a stronger justification for the statistical methods employed, particularly the use of the Bonferroni correction and the choice of p<0.007p < 0.007p<0.007 as the significance threshold. Additionally, we recommend expanding the discussion to address the limitations inherent to the retrospective study design and to contextualize your findings within the existing literature.
Below, we outline specific comments and suggestions aimed at enhancing the clarity, rigor, and overall impact of your manuscript.
Review of the Manuscript
Strengths of the Study
- Innovative Approach: The study explores two-port laparoscopic-assisted ovariohysterectomy in small-sized dogs, a relevant and less-studied area.
- Well-documented Methods: The surgical procedure and postoperative management are described in detail, making the methods reproducible.
- Clinical Relevance: The findings suggest that the method is safe and effective, with minimal complications, which holds promise for veterinary surgical practices.
- Clear Data Presentation: The use of tables and figures effectively summarizes key results, enhancing the manuscript's readability.
Weaknesses of the Study
- Justification of Statistical Threshold:
- The use of p<0.007 as the significance threshold following Bonferroni correction is not explained adequately. The manuscript does not discuss how this impacts the study's statistical power or type II error risks.
- Results with p-values between 0.007 and 0.05 are not discussed, potentially overlooking meaningful trends.
- Retrospective Design Limitations:
- The study acknowledges its retrospective nature but does not discuss potential biases or limitations in detail, such as the selection criteria for the cases included.
- Sample Size Constraints:
- With 77 cases, the study’s power to detect small but clinically significant differences may be limited, especially under a stringent significance threshold.
- Minor Formatting and Structural Issues:
- The introduction could better contextualize the study’s novelty by referencing existing literature on laparoscopic surgery in small-sized dogs.
- The discussion could benefit from a more thorough exploration of the biological and clinical implications of the findings.
Recommendations for Revision
1. Major Revisions
- Statistical Analysis: Provide a justification for the Bonferroni correction and the choice of p<0.007. Discuss results with p-values between 0.007 and 0.05 as trends, if appropriate.
- Limitations: Expand on the limitations of the retrospective design and discuss how they might impact the conclusions.
2. Minor Revisions
- Discussion: Deepen the interpretation of the findings, particularly the clinical relevance and implications for broader populations.
Overall Recommendation: Major Revision
While the study is innovative and addresses a clinically relevant topic, the lack of justification for the statistical methods and the limited discussion of the study’s limitations require significant revisions to enhance its scientific rigor and interpretability.
Best Regards
Author Response
Letter to the reviewer
Justification of Statistical Threshold:
The use of p<0.007 as the significance threshold following Bonferroni correction is not explained adequately. The manuscript does not discuss how this impacts the study's statistical power or type II error risks.
Author’s response: Bonferroni correction was used to reduce the risk of a type I error for multiple comparisons. We included this explanation into the revised manuscript.
Line 168-169: Bonferroni correction was applied, with the significance level set at p < 0.007, to reduce the risk of a type I error.
Results with p-values between 0.007 and 0.05 are not discussed, potentially overlooking meaningful trends.
Author’s response: The p-values are either more than 0.05 (not significant anyway) or less than 0.007 (significant even with stricter alpha) except one (p value=0.024) for age comparison between two groups. Thus, we concluded that the risk of a type II error is minimal.
Retrospective Design Limitations:
The study acknowledges its retrospective nature but does not discuss potential biases or limitations in detail, such as the selection criteria for the cases included.
Author’s response: We added more details in the limitation.
Line 321-327: The limitations of this study include its retrospective nature. Only dogs with mild pyometra were considered for this surgery. Those with perforations, peritonitis, or significant uterine distension were excluded. In such cases, open ovariohysterectomy should be indicated.
Sample Size Constraints:
With 77 cases, the study’s power to detect small but clinically significant differences may be limited, especially under a stringent significance threshold.
Author’s response: The study limitations
Line 327-329: The retrospective nature of the study and its limited sample size necessitate further investigation. Future studies with larger case numbers are warranted to investigate the clinical outcomes of two-port laparoscopic-assisted ovariohysterectomy for mild to moderate hydrometra and pyometra.
Minor Formatting and Structural Issues:
The introduction could better contextualize the study’s novelty by referencing existing literature on laparoscopic surgery in small-sized dogs.
Author’s response: Thank you for your suggestion. The authors conducted thorough literature review; However, to the best of our knowledge, literature focused on laparoscopic surgery in small-sized dogs is limited. We have included as many references as possible if appropriate in the manuscript.
The discussion could benefit from a more thorough exploration of the biological and clinical implications of the findings.
Author’s response: Those with significant uterine distension or peritonitis, including severe adhesion around abdominal organs, are not suitable for two-port laparoscopic-assisted ovariohysterectomy, and this surgery is indicated only for mild to moderate hydrometra and pyometra.
Line322-324
Recommendations for Revision
1. Major Revisions
Statistical Analysis: Provide a justification for the Bonferroni correction and the choice of p<0.007. Discuss results with p-values between 0.007 and 0.05 as trends, if appropriate.
Author’s response: Bonferroni correction was used to reduce the risk of a type I error for multiple comparisons. We included this explanation into the revised manuscript.
Line 168-169: Bonferroni correction was applied, with the significance level set at p < 0.007, to reduce the risk of a type I error.
The p-values are either more than 0.05 (not significant anyway) or less than 0.007 (significant even with stricter alpha) except one (p value=0.024) for age comparison between two groups. Thus, we concluded that the risk of a type II error is minimal.
Limitations: Expand on the limitations of the retrospective design and discuss how they might impact the conclusions.
Line 321-324: The limitations of this study include its retrospective nature. Only dogs with mild pyometra were considered for this surgery. Those with perforations, peritonitis, or significant uterine distension were excluded. In such cases, open ovariohysterectomy should be indicated.
2. Minor Revisions
Discussion: Deepen the interpretation of the findings, particularly the clinical relevance and implications for broader populations.
Author’s response: Those with perforations, peritonitis, or significant uterine distension are not suitable for this surgery. In such cases, open ovariohysterectomy should be indicated. This surgery is indicated only for mild to moderate hydrometra and pyometra. The retrospective nature of the study and its limited sample size necessitate further investigation. Future studies with larger case numbers are warranted to investigate the clinical outcomes of two-port laparoscopic-assisted ovariohysterectomy for hydrometra and pyometra.
